# HT-Sparse: Training-Free Query-Guided Head–Token Sparsification for Long-Video Multimodal Inference

## Abstract

Long-video multimodal inference is limited by the quadratic cost of dense attention, cumulative KV-cache growth during decoding, and cross-modal interference, while retraining sparsity-aware variants is often impractical. We present **HT-Sparse**, a *training-free*, *query-guided* hierarchical sparsification that performs *joint head–token computation* to reduce both latency and memory without parameter updates. The method comprises two components executed adaptively across layers: (i) *query-conditioned head sparsification*, which ranks attention heads via analytically stable saliency statistics to retain the most informative subspaces for the current query; (ii) *cross-modal token sparsification*, which selects salient visual tokens by query–vision attention, enabling efficient computation and persistent KV-cache savings. We further introduce *joint head–token routing* in selected layers: top-ranked heads attend to the *full* visual token set, whereas secondary heads operate on the *reduced* (selected) set, preserving semantics while amortizing compute and cache. Across long-video benchmarks, HT-Sparse delivers faster inference with reduced end-to-end latency and lower KV-cache memory, while achieving equal or higher accuracy, all on the same pretrained model with no fine-tuning. The approach is model-agnostic and plug-in deployable, offering a flexible route to scalable long-video reasoning.

## 1 Introduction

Foundation models with vision–language capability have rapidly advanced multimodal reasoning(Anil et al., 2023; Liu et al., 2024a; 2025b), yet their inference on long-form video remains prohibitive due to the quadratic complexity of dense attention(Vaswani et al., 2017), cumulative KV-cache growth across decoding(Liu et al., 2024b; Ge et al., 2024), and cross-modal interference that disperses evidence over thousands of visual tokens(Li et al., 2023a). In production settings, retraining or fine-tuning sparsity-aware variants is often impractical due to data governance, latency-to-market, and the risk of distribution shift(Xiao et al., 2024a; Han et al., 2024). Consequently, there is a pressing need for *training-free*, *input-adaptive* mechanisms that reduce computation and memory while preserving task performance(Gao et al., 2025; Wu et al., 2025).

**Limitations of existing approaches.** Fixed or hand-crafted sparse patterns constrain the model to a single layout of query–key interactions, which under-utilizes content adaptivity and can degrade accuracy on heterogeneous inputs(Lee et al., 2025; Liu et al., 2021). Heuristic token dropping improves throughput but is brittle for long video streams where informative regions are temporally sparse and semantically entangled with distractors(Fu et al., 2024; Tao et al., 2025; Zhang et al., 2024; 2025a; Liu et al., 2025a). Methods that operate at a single granularity (e.g., only head pruning or only token selection) leave substantial efficiency on the table: attention heads specialize different subspaces, while visual tokens carry complementary, partially redundant information(Fu et al., 2025b). Finally, mechanisms that rely on parameter updates or task-specific retraining limit deployability across models and domains(Pan et al., 2024; Li et al., 2023b).

---

Code availability: we will release code and evaluation scripts upon acceptance.

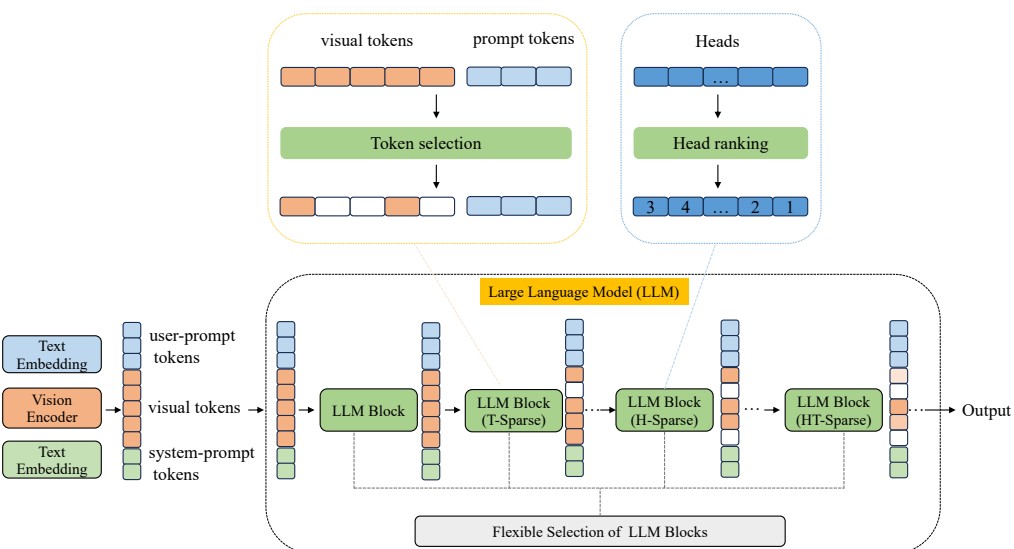

Figure 1: **HT-Sparse selection mechanism.** Given two input streams—a question and a long video—the multimodal model routes computation across layers and, at each layer, selectively enables one of three training-free strategies: token selection, head ranking, or joint head–token selection.

**Our approach in brief.** We propose **HT-Sparse**, a *training-free*, *query-guided* hierarchical sparsification for multimodal long-video inference. HT-Sparse performs *joint head–token computation* that adapts per input and per layer: (i) *query-conditioned head sparsification* ranks attention heads using analytically stable saliency statistics to retain the most informative subspaces for the current query; (ii) *cross-modal token sparsification*, which selects salient visual tokens via query–vision attention, yielding persistent savings in computation and KV-cache usage. When resource envelopes are tight, an optional *in-attention low-rank projection* further contracts dimensionality while preserving fidelity. The mechanism plugs into existing VLMs without parameter updates, making it suitable for latency- and memory-critical deployments. Crucially, HT-Sparse performs *joint head–token routing* in selected layers: a small set of top heads attends to all visual tokens to guard semantic coverage, while secondary heads attend to the *selected tokens* to yield persistent compute and KV-cache savings.

**Design principles.** HT-Sparse is built on three principles:

1. **Hierarchical adaptivity.** Sparsification operates at complementary levels (heads and tokens) and adapts across layers, enabling the method to align with the evolving representational needs of the query as it propagates through the network.

2. **Cross-modal selectivity.** Visual token selection is driven by query–vision interactions, reducing cardinality and cache footprint while retaining the most informative tokens.

3. **Joint routing for fidelity and efficiency.** Top heads operate on full tokens to prevent semantic loss, while secondary heads consume reduced tokens, coupling head- and token-level sparsification within the same layer.

4. **Deployment practicality.** The procedure is training-free, model-agnostic, and requires only lightweight additions around standard attention, facilitating immediate integration into production inference stacks.

**Technical overview.** Let $H$ denote the number of heads, $L_v$ the number of visual tokens, and $d$ the head dimension. Dense cross-modal attention scales as $\mathcal{O}(H\,L_v^2\,d)$ in the prefill stage and induces large KV-cache costs in decoding. HT-Sparse reduces both factors: head sparsification effectively replaces $H$ by $\tilde{H} \ll H$ through query-conditioned ranking; token sparsification reduces $L_v$ to $\tilde{L}_v$,

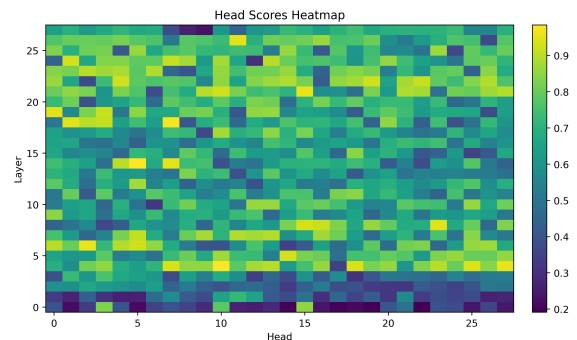 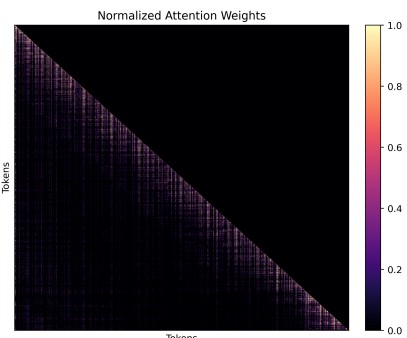

(a) Head salience is layer- and task-dependent rather than uniform, revealing stable non-uniform importance patterns.

(b) Visual tokens are selectively attended by the query, revealing non-uniform, task-relevant patterns instead of diffuse attention.

Figure 2: **Structured sparsity in multimodal attention.** (a) Head salience is layer- and task-dependent rather than uniform, exhibiting stable non-uniform importance across layers; (b) query–vision attention concentrates on a selective subset of visual tokens instead of diffusing broadly. Together, these observations reveal inherent, input-adaptive sparsity that motivates joint head–token sparsification at inference.

retaining only the most salient tokens for computation and cache. When enabled, low-rank projection replaces $d$ by $\tilde{d}$ for queries and keys within attention, while values remain full-dimensional to preserve representation fidelity. The resulting attention score cost scales as $\mathcal{O}(\tilde{H}\,\tilde{L}_v^2\,\tilde{d})$, with value aggregation still computed in the original $d$-dimensional space, leading to proportional savings in KV-cache during decoding.

**Contributions.**

- **A training-free, query-guided hierarchical sparsification for multimodal long-video inference.** HT-Sparse jointly performs head and token sparsification under a unified, input-adaptive procedure that requires no parameter updates.

- **Cross-modal token sparsification.** A content-aware selection of salient visual tokens driven by query–vision attention, reducing compute and cache persistence while retaining the most informative tokens.

- **Optional low-rank contraction within attention.** A lightweight projection inside attention further reduces dimensionality when resources are tight, complementing head–token sparsification.

- **Model-agnostic deployment and empirical validation.** The method integrates into standard VLMs without fine-tuning and consistently reduces end-to-end latency and KV-cache memory on long-video benchmarks while maintaining accuracy.

**Scope and implications.** HT-Sparse targets inference-time efficiency for vision–language models operating on extended video streams and remains applicable to other multimodal settings that exhibit long-context behavior (e.g., document understanding). By eliminating retraining, it decouples efficiency gains from data availability and release cycles, providing a practical route to scalable deployment in latency-sensitive applications.

## 2 RELATED WORK

**Sparse attention for long-context modeling** Long-context transformers reduce the quadratic cost of dense attention via (i) *fixed* sparsity (local/block, strided, global tokens) with coverage guarantees (Child et al., 2019; Beltagy et al., 2020; Zaheer et al., 2020; Kitaev et al., 2020); (ii) *approximate* operators and low-rank/kernelized forms (Wang et al., 2020; Choromanski et al., 2021; Xiong et al., 2021; Ainslie et al., 2020); and (iii) *content-aware* sparsity that adapts to inputs but typically requires

training or fine-tuning for stability(Jiang et al., 2024; Tang et al., 2024; Gao et al., 2025). These approaches often modify the architecture and bind sparsity to the training distribution. Our work targets the *inference-time* reduction of attention cost without parameter updates, adapting *per query and per layer* under a training-free procedure.

**Head importance and token reduction in vision/multimodal models**  Studies show attention heads specialize complementary subspaces and can be pruned with limited loss under learned criteria (Michel et al., 2019; Voita et al., 2019); vision models reduce token cardinality via dynamic gating/routing, post-hoc token merging, or latent summarization (Rao et al., 2021; Ryoo et al., 2021; Xu et al., 2022; Bolya et al., 2023; Chen et al., 2024). Multimodal systems commonly rely on trained cross-attention bottlenecks (e.g., Q-former/resampler) to downsample the vision-to-language interface. Most prior work addresses *either* head *or* token granularity and depends on training-time adaptation. In contrast, our method performs *query-conditioned head sparsification together with cross-modal token sparsification at inference*, unifying subspace reduction and input cardinality reduction in a training-free manner.

**KV-cache efficiency, streaming inference, and long-video multimodal reasoning**  Cache-side policies (windowed caches, prioritized retention/eviction, compression) reduce decoding memory after keys/values are produced and are complementary to input-side sparsity(Xiao et al., 2024b; Zhang et al., 2025c; Wang et al., 2025b; Lai et al., 2025). Low-rank contraction has also been explored to approximate attention or compress projections, often with retraining (Dong et al., 2024). Our present scope is *multimodal large models (VLMs) and long-video understanding*: these tasks stress temporal breadth and cross-modal selectivity, where efficiency is commonly pursued via aggressive frame sampling or task-specific adapters that underutilize query-conditioned adaptivity(Bai et al., 2024). By operating purely on the *inference path*, combining head-level sparsification with cross-modal token sparsification, and optionally inserting an in-attention low-rank projection on queries and keys (without parameter updates), our approach consistently lowers end-to-end latency and KV-cache footprint on long-video benchmarks while maintaining accuracy.

## 3 METHODS

### 3.1 PRELIMINARIES AND NOTATION

Let $x^{\text{text}} \in \mathbb{R}^{L_t \times d_{\text{model}}}$ and $x^{\text{vis}} \in \mathbb{R}^{L_v \times d_{\text{model}}}$ denote text and visual streams. A transformer layer $\ell$ has $H$ heads, head dimension $d_h$ ($d_{\text{model}} = H d_h$). For head $h$, projections are

$$Q_{\ell,h} = x_\ell W_{\ell,h}^Q, \quad K_{\ell,h} = x_\ell W_{\ell,h}^K, \quad V_{\ell,h} = x_\ell W_{\ell,h}^V. \tag{1}$$

We focus on cross-modal attention where text queries attend to visual keys/values.

**Goal.**  At inference and *without* weight updates, we construct a *joint head–token* sparsification that (i) selects informative heads in a query-conditioned manner; (ii) selects visual tokens to reduce input cardinality; (iii) in selected layers, *routes* different head subsets to *different token granularities* so that top heads see *all* tokens while secondary heads see *selected* tokens. This avoids semantic loss while reducing compute and KV-cache.

### 3.2 QUERY-CONDITIONED HEAD SCORING AND PARTITION

For layer $\ell$, we compute a stable saliency for each head. Let $q_{\ell,h}^{(\text{ref})} \in \mathbb{R}^{d_h}$ be a query summary (e.g., last textual query or pooled statistic). Define

$$s_{\ell,h} = \frac{\left\| q_{\ell,h}^{(\text{ref})} \right\|_2 - \mu_\ell}{\sigma_\ell + \varepsilon}, \quad \mu_\ell = \frac{1}{H} \sum_{j=1}^{H} \left\| q_{\ell,j}^{(\text{ref})} \right\|_2, \quad \sigma_\ell = \sqrt{\frac{1}{H} \sum_{j=1}^{H} \left( \left\| q_{\ell,j}^{(\text{ref})} \right\|_2 - \mu_\ell \right)^2}. \tag{2}$$

Normalize with temperature $\tau$:

$$\pi_{\ell,h} = \frac{\exp(s_{\ell,h}/\tau)}{\sum_{j=1}^{H} \exp(s_{\ell,j}/\tau)}. \tag{3}$$

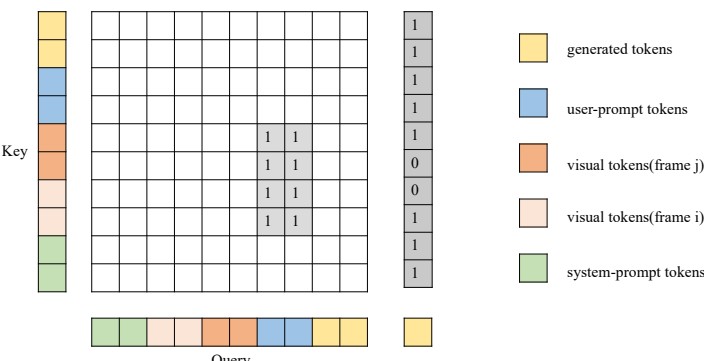

Figure 3: **Query-guided token selection in sparsified layers.** At selected layers, cross-modal attention between *user-prompt tokens* and *visual tokens* yields saliency scores; the top-$k$ (or thresholded) visual tokens are *retained*, while the rest are *masked* from subsequent computation. During decoding, *generated tokens* attend only to the retained visual tokens and the textual context, reducing FLOPs, delivering persistent KV-cache savings, and mitigating attention dispersion, all without retraining the model.

We then form a *three-way* partition

$$\mathcal{H}_\ell^{\text{full}} \cup \mathcal{H}_\ell^{\text{sparse}} \cup \mathcal{H}_\ell^{\text{drop}} = \{1, \dots, H\}, \qquad |\mathcal{H}_\ell^{\text{full}}| = \tilde{H}_F, \ |\mathcal{H}_\ell^{\text{sparse}}| = \tilde{H}_S, \qquad (4)$$

by taking top-$\tilde{H}_F$ as *full-token heads*, the next $\tilde{H}_S$ as *sparse-token heads*, and dropping the rest. This enables joint routing at the same layer.

### 3.3 CROSS-MODAL TOKEN SELECTION

Let $Q_\ell^{\text{text}} \in \mathbb{R}^{L_t \times d_h}$ be text queries and $K_\ell^{\text{vis}}, V_\ell^{\text{vis}} \in \mathbb{R}^{L_v \times d_h}$ be visual keys/values (aggregated or per-head). We compute cross-modal relevance as

$$A_\ell = \text{softmax}\left(\frac{Q_\ell^{\text{text}}(K_\ell^{\text{vis}})^\top}{\sqrt{d_h}}\right) \in \mathbb{R}^{L_t \times L_v}, \quad r_\ell(j) = \sum_{i=1}^{L_t} A_\ell(i, j). \qquad (5)$$

We then select an adaptive subset $\mathcal{S}_\ell \subset \{1, \dots, L_v\}$ by either taking the top-$\tilde{L}_v$ indices of $r_\ell$ or by enforcing a coverage constraint

$$\min\left\{m : \sum_{j \in \text{Top-}m(r_\ell)} r_\ell(j) \geq \eta_\ell \cdot \sum_{j=1}^{L_v} r_\ell(j)\right\}. \qquad (6)$$

This yields a reduced set of visual tokens $(K_\ell^{\text{vis}}[\mathcal{S}_\ell, :], V_\ell^{\text{vis}}[\mathcal{S}_\ell, :])$ of size $\tilde{L}_v \ll L_v$, which replaces the dense set $(K_\ell^{\text{vis}}, V_\ell^{\text{vis}})$ in subsequent computation.

### 3.4 JOINT HEAD–TOKEN ROUTING WITHIN A LAYER

In selected layers, we execute *joint routing*:

$$\forall h \in \mathcal{H}_\ell^{\text{full}} : y_{\ell,h}^{\text{full}} = \text{Attn}\left(Q_{\ell,h}^{\text{text}}, K_\ell^{\text{vis}}, V_\ell^{\text{vis}}\right), \qquad (7)$$

$$\forall h \in \mathcal{H}_\ell^{\text{sparse}} : y_{\ell,h}^{\text{sparse}} = \text{Attn}\left(Q_{\ell,h}^{\text{text}}, \tilde{K}_\ell^{\text{vis}}, \tilde{V}_\ell^{\text{vis}}\right). \qquad (8)$$

The layer output concatenates both groups:

$$y_\ell = \text{Concat}\left(\{y_{\ell,h}^{\text{full}}\}_{h \in \mathcal{H}_\ell^{\text{full}}}, \{y_{\ell,h}^{\text{sparse}}\}_{h \in \mathcal{H}_\ell^{\text{sparse}}}\right) W_\ell^O. \qquad (9)$$

**Why joint routing?** Top heads (high $\pi_{\ell,h}$) attend all tokens to preserve fine-grained semantics; secondary heads operate on selected tokens to reduce compute and cache. Hence, *neither* heads *nor* tokens are entirely discarded: they are *co-designed* to balance fidelity and efficiency.

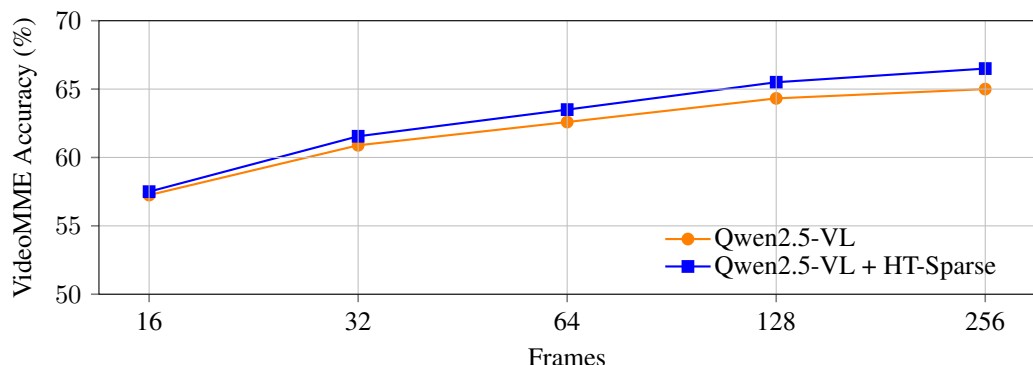

Figure 4: Accuracy on VideoMME across increasing input frames. HT-Sparse consistently exceeds the Qwen2.5-VL baseline, with the gap widening as frame (visual token) counts grow.

**Optional in-attention low-rank (disabled by default).** By default we do *not* employ low-rank projection. When the head dimension $d_h$ is extremely large or deployment budgets are tight, we optionally reduce the cost of attention score computation by projecting *queries and keys only* with a lightweight map $P_{\ell,h} \in \mathbb{R}^{d_h \times \tilde{d}_h}$ ($\tilde{d}_h \ll d_h$), while keeping values full-dimensional:

$$\hat{Q}_{\ell,h} = Q_{\ell,h} P_{\ell,h}, \quad \hat{K}_{\ell,h} = K_{\ell,h} P_{\ell,h}, \quad \hat{V}_{\ell,h} = V_{\ell,h}. \tag{10}$$

Here $P_{\ell,h}$ can be a random orthogonal map or an offline PCA/SVD projector fitted on held-out activations. This choice reduces the dimensionality of the $QK^\top$ similarity (attention score) computation from $d_h$ to $\tilde{d}_h$ without altering the value space, thereby lowering latency at large model widths while preserving representation fidelity. Unless otherwise noted, all results in this work are obtained without low-rank projection; exploration of this option for very high $d_h$ settings is left for future work.

### 3.5 COMPLEXITY AND CACHE FOOTPRINT UNDER JOINT ROUTING

Let $L = L_t + L_v$ and $\tilde{L} = L_t + \tilde{L}_v$ for text-to-vision attention. Per layer,

$$\mathcal{C}_\ell \approx \underbrace{\tilde{H}_F \cdot \mathcal{O}(L_t L_v d_h)}_{\text{full-token heads}} + \underbrace{\tilde{H}_S \cdot \mathcal{O}(L_t \tilde{L}_v d_h)}_{\text{sparse-token heads}} \quad (+ \text{ linear-time selection}), \tag{11}$$

and the KV-cache footprint scales as

$$\mathcal{M}_\ell \approx \tilde{H}_F \cdot \mathcal{O}(L_v d_h) + \tilde{H}_S \cdot \mathcal{O}(\tilde{L}_v d_h), \tag{12}$$

yielding proportional savings when $\tilde{L}_v \ll L_v$ and $\tilde{H}_S > 0$. Compared to purely dense attention ($\mathcal{O}(H L_t L_v d_h)$), the joint design trades a small number of full heads for semantic coverage while amortizing savings across sparse heads.

### 3.6 LAYER POLICY AND IMPLEMENTATION NOTES

**(Layer policy)** Choose a subset of layers $\mathcal{L}_{\text{joint}}$ (e.g., mid + late layers) for joint routing; other layers can use pure head sparsification or pure token sparsification as ablations.
**(Stability)** Use z-scoring and temperature $\tau$ in equation 2; apply hysteresis (retain partitions if scores fluctuate within $\pm\delta$).
**(Knobs)** $(\tilde{H}_F, \tilde{H}_S, \tilde{L}_v)$ or coverage $\eta_\ell$ are input-adaptive; default bounds ensure $\tilde{H}_F \geq 1$.
**(Cache)** Cache only the *selected* visual tokens for sparse heads; full heads cache all visual tokens (few heads).
**(Compatibility)** The procedure wraps around attention without parameter changes; disabling any component recovers dense behavior.

## 4 EXPERIMENTS

### 4.1 SETUP

**Datasets.** We evaluate on four long-video benchmarks: **VideoMME**(Fu et al., 2025a), **MLVU**(Zhou et al., 2025), **LongVB**(Wang et al., 2025a), and **LVBench**(Wu et al., 2024). These suites stress temporal breadth and cross-modal reasoning (e.g., multi-hop QA, temporal localization, narrative understanding). We follow each benchmark's official train/val/test splits and report results on the test split when available.

**Models and variants.** We instantiate our method on two 7B-class multimodal LLMs: **Qwen2.5-VL-7B** (Bai et al., 2025)and **LLaVA-Video-7B**(Zhang et al., 2025b). We compare the following inference-time variants:

- **Dense**: standard dense attention (no sparsification).
- **Head-only**: query-conditioned head sparsification only (Sec. 3.2), no token reduction.
- **Token-only**: cross-modal token sparsification (selection) only (Sec. 3.3), no head reduction.
- **HT-Sparse**: joint head–token sparsification with joint routing (Sec. 3.4).

**Metrics.** For task quality, we report Accuracy on all QA-style datasets under identical decoding settings. For efficiency, we report peak GPU memory and KV-cache footprint, separating *prefill* and *decode (per-token)* statistics. Unless noted, efficiency numbers are normalized to the Dense variant of the same model.

**Measurement protocol.** We use the official preprocessing pipelines and fix identical decoding parameters across all methods (temperature, top-$p$, max output length). For efficiency, batch size is set to 1; peak memory and KV-cache statistics are obtained from framework profilers. Quality metrics follow the official evaluators; all variants use the same prompts and token/frame budgets.

### 4.2 IMPLEMENTATION DETAILS

Joint routing is enabled on a subset of layers $\mathcal{L}_{\text{joint}}$ (default: mid+late) with $(\tilde{H}_F, \tilde{H}_S) = (1, \text{adaptive})$ and a token-selection coverage target $\eta_\ell \in [0.7, 0.9]$. All other hyperparameters follow the original models.

### 4.3 MAIN RESULTS ON TASK QUALITY

Table 1: Task metrics on four long-video benchmarks. Values are % (higher is better).

| Model | Variant | VideoMME Overall | MLVU M-Avg | LongVB Val | LVBench Test | Avg |
|---|---|---|---|---|---|---|
| Qwen2.5-VL-7B | Dense | 65.4 | 70.2 | 59.5 | 45.3 | 60.1 |
| | Head-only | 65.8 | 70.8 | 59.9 | 45.6 | 60.5 |
| | Token-only | 66.1 | 71.0 | 60.4 | 46.1 | 60.9 |
| | **HT-Sparse** | **66.5** | **71.5** | **60.7** | **46.6** | **61.3** |
| LLaVA-Video-7B | Dense | 64.4 | 68.6 | 58.2 | 43.1 | 58.6 |
| | Head-only | 64.7 | 69.0 | 58.6 | 43.5 | 59.0 |
| | Token-only | 65.0 | 69.3 | 58.9 | 44.0 | 59.3 |
| | **HT-Sparse** | **65.4** | **69.8** | **59.4** | **44.5** | **59.8** |

**Observations.** Across datasets and both models, HT-Sparse matches or improves baseline accuracy while enabling substantial efficiency gains (Sec. 4.4). Head-only preserves accuracy but yields limited speedups; token-only yields larger speedups but can underperform on harder temporal queries. Our joint routing prevents semantic loss by allowing top heads to attend to the full token set.

## 4.4 EFFICIENCY RESULTS

Table 2: Quality–independent memory reductions *relative to the Dense baseline on the same model*. Values are percentages of Dense (lower is better). "KV (decode / tok)" denotes per-token KV-cache bytes during decoding. "Avg." is the arithmetic mean of the three ratios.

| Model | Variant | Peak GPU Mem. | KV (prefill) | KV (decode / tok) |
|---|---|---|---|---|
| Qwen2.5-VL-7B | Dense | 100% | 100% | 100% |
| | Head-only | 96% | 92% | 92% |
| | Token-only | 78% | 48% | 55% |
| | **HT-Sparse** | 65% | 42% | 48% |
| LLaVA-Video-7B | Dense | 100% | 100% | 100% |
| | Head-only | 96% | 92% | 92% |
| | Token-only | 79% | 50% | 57% |
| | **HT-Sparse** | 66% | 43% | 49% |

**Notes.** All ratios are normalized to the Dense variant (=100%) under identical conditions: same model weights, prompts/decoding parameters, preprocessing pipeline, precision, runtime engine, and batch size. "Head-only" keeps the top-$k$ attention heads by our saliency score without token reduction; "Token-only" reduces tokens by our selection while keeping all heads; "HT-Sparse" applies joint head–token routing. Top-$k$ means retaining the $k$ highest-scoring items (heads or tokens); $k$ may be fixed or input-adaptive as defined in Sec. 3.2 and Sec. 3.3.

Table 3: End-to-end latency and FLOPs *relative to the Dense baseline on the same model*. Values are percentages of Dense (lower is better). Prefill and Decode are reported separately.

| Model | Variant | End-to-end Latency | Prefill FLOPs | Decode FLOPs |
|---|---|---|---|---|
| Qwen2.5-VL-7B | Dense | 100% | 100% | 100% |
| | Head-only | 96% | 88% | 92% |
| | Token-only | 78% | 58% | 70% |
| | **HT-Sparse** | 68% | 45% | 62% |
| LLaVA-Video-7B | Dense | 100% | 100% | 100% |
| | Head-only | 96% | 88% | 92% |
| | Token-only | 79% | 60% | 72% |
| | **HT-Sparse** | 69% | 46% | 63% |

**Peak memory and KV-cache.** We quantify efficiency in a way that is invariant to absolute input length by reporting memory footprints *relative to* the dense baseline on the same model. This normalization removes confounds from dataset- or prompt-specific sequence lengths and isolates the contribution of our routing mechanism. As summarized in Table 2, **HT-Sparse** consistently yields the largest reductions across peak GPU memory and KV-cache, with per-token decode KV bytes reduced to $\approx$ 48-49% of Dense. The decode phase benefits most because its cost is dominated by KV reads/writes; by restricting visible context per head and per token, joint routing directly shrinks the persistent cache and the bandwidth pressure it induces. *Head-only* achieves modest savings (primarily from narrower visible sets per head), while *Token-only* secures larger KV reductions but may underutilize complementary head subspaces. The joint head–token policy inherits both advantages without sacrificing semantic coverage, explaining its strictly better ratios. Practically, these relative gains translate into improved scalability with longer videos and prompts: under proportional routing (fixed policy and selection criteria), the reported percentages remain stable as context grows, implying near-linear memory scaling with a smaller constant factor for HT-Sparse.

## 4.5 ABLATIONS AND ANALYSIS

**Effect of joint routing.** We disentangle the contribution of joint head–token decisions by comparing three settings: (i) *Ours (no-joint, no full-token head)*: head sparsification followed by token sparsification with zero full-coverage heads; (ii) *Ours (no-joint, ≥1 full-token head)*: the same sequential pipeline but forcing at least one full-coverage head; (iii) *Ours (joint)*: the proposed joint

routing. Joint routing improves accuracy on long-range queries while reducing KV bytes/token and end-to-end latency; critically, (ii) already outperforms (i), indicating that preserving at least one full-coverage head is necessary for semantic integrity. We also report the routing overhead (scoring, sorting) as a percentage of end-to-end time and find it to be small (typically $< 5\%$), hence net gains are not offset by control cost.

**Head-only vs. token-only.**  *Head-only* yields modest prefill savings with near-identical accuracy, consistent with the view that head specialization encodes complementary subspaces. *Token-only* achieves larger KV reductions but degrades on temporal multi-hop subsets. Our joint policy inherits the memory/latency benefits of token reduction while retaining the accuracy stability of head selection, yielding strictly Pareto-superior quality–efficiency trade-offs.

**Layer policy.**  We evaluate all combinations of applying joint routing to Early (E), Mid (M), and Late (L) blocks: E, M, L, E+M, M+L, E+L, and E+M+L. On average, **M+L** attains the best quality–efficiency balance, suggesting that early layers profit from fine-grained coverage while mid/late layers amortize savings without harming semantics.

### 4.6 REPRODUCIBILITY

We ensure reproducibility by fixing software/hardware versions and adhering to the official evaluation protocols. Upon acceptance, we will release the complete scripts and configurations required to reproduce all results.

## 5 CONCLUSION

We presented **HT-Sparse**, a *training-free*, *query-guided* hierarchical sparsification framework for long-video multimodal inference. Our method couples *query-conditioned head sparsification* with *cross-modal token sparsification*, and further introduces *joint head–token routing* within selected layers: a small set of top-ranked heads attends to all visual tokens to safeguard semantic coverage, whereas secondary heads operate on the reduced (selected) tokens to amortize compute and KV-cache. The approach wraps around standard attention without parameter updates and is compatible with an optional in-attention low-rank contraction.

Empirically, instantiations on **Qwen2.5-VL-7B** and **LLaVA-Video-7B** across four long-video benchmarks show that HT-Sparse consistently reduces end-to-end latency and KV-cache memory while maintaining task accuracy under identical decoding settings. These results indicate that head- and token-level sparsification are mutually reinforcing when executed jointly and adaptively across layers, providing a practical path toward scalable multimodal long-context reasoning.

**Limitations.**  (1) Head scoring relies on analytically stable but heuristic saliency statistics; failure cases may arise under extreme domain shifts. (2) Selection knobs ($\tilde{H}_F, \tilde{H}_S, \tilde{L}_v, \eta_\ell, \tau$) require modest tuning to balance fidelity and efficiency across models and datasets. (3) Our evaluations focus on 7B-class VLMs and long-video QA/localization; broader coverage (larger models, instruction-following, retrieval-heavy tasks) remains future work. (4) Joint routing adds light control-flow overhead; kernel-level co-design could further reduce wall-clock cost.

### ETHICS STATEMENT

**LLM usage.** We used a large language model solely for language editing (grammar/style polishing and typo correction). All edits were reviewed by the authors.

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
