# OpenReview forum: "HT-Sparse: Training-Free Query-Guided Head–Token Sparsification for Long-Video Multimodal Inference"
_ICLR.cc/2026/Conference — Submitted to ICLR 2026_

### Official Review · Reviewer_egSG · 2025-10-18

**Soundness:** 2
**Presentation:** 2
**Contribution:** 2
**Rating:** 2
**Confidence:** 4

**Summary:**

This paper proposes a training-free, query-guided hierarchical sparsification method, reduces latency and memory via joint head–token computation without parameter updates. It has two adaptive components: query-conditioned head sparsification (ranks heads to retain key subspaces) and cross-modal token sparsification (selects salient visual tokens for KV-cache savings). It also adds joint head–token routing in some layers (top heads use full tokens, secondary heads use reduced tokens). Tested on long-video benchmarks, HT-Sparse boosts inference speed, cuts end-to-end latency and KV-cache memory, maintains or improves accuracy—all on pretrained models without fine-tuning. It is model-agnostic, plug-and-play, and enables scalable long-video reasoning.

**Strengths:**

1. This paper explores how to enhance the efficiency of long-video understanding through cross-modal attention sparsification in a training-free manner, while achieving a certain degree of performance improvement compared to the baseline.

2. This paper addresses attention sparsification from both the head-level and token-level perspectives, and verifies the effectiveness of the combination of these two approaches.

**Weaknesses:**

1. The elaboration on the core contributions and innovations of this paper is insufficient. Most of the contributions mentioned in this paper, such as training-free and cross-modality, have been addressed in numerous previous works [1][2][3]. Compared with baseline methods (token-only or head-only), the improvement presented is relatively trivial. It is recommended that the authors supplement the discussion with previous works to strengthen the contributions that are exclusive to this study.

2. The writing of the method section in this paper seems overly concise, making it difficult to read and lacking detailed descriptions of method details.

3. The ablation study section of this paper is not sufficiently comprehensive and solid. For instance, it lacks the analysis of HT-Sparse's performance variation under different values of k. Additionally, there is no investigation into the method's performance impact on image tasks. From the reviewer's perspective, this method does not include specialized designs tailored for video tasks; thus, merely evaluating it on a few video benchmarks is insufficient.

[1] AdaReTaKe: Adaptive Redundancy Reduction to Perceive Longer for Video-language Understanding

[2] DyCoke: Dynamic Compression of Tokens for Fast Video Large Language Models

[3] FastVID: Dynamic Density Pruning for Fast Video Large Language Models

**Questions:**

My primary concern lies with the method’s core contributions and the scope of its experimental work; I believe both aspects fail to meet the acceptance criteria of the conference.

---

### Official Review · Reviewer_9LLf · 2025-10-23

**Soundness:** 1
**Presentation:** 3
**Contribution:** 2
**Rating:** 4
**Confidence:** 3

**Summary:**

This paper proposes HT-Sparse, a training-free method to tackle challenges in long-video multimodal inference, such as high attention costs, KV-cache growth, and cross-modal interference. HT-Sparse combines query-conditioned head sparsification, which selects the most informative attention heads, and cross-modal token sparsification, which reduces visual tokens based on query relevance. It also introduces joint head–token routing to balance computational efficiency and semantic preservation. Tested on long-video benchmarks, HT-Sparse achieves faster inference, lower memory usage, and comparable or better accuracy, all without retraining or fine-tuning, making it a scalable and plug-and-play solution for multimodal reasoning.

**Strengths:**

1. The paper is well-written, with the methodology and experimental results clearly and systematically presented.
2. The research topic is highly practical, as developing a training-free method that effectively reduces inference memory usage and latency offers significant benefits for real-world deployment of models.
3. Without requiring model retraining, HT-Sparse achieves impressive performance on several long-video benchmarks, maintaining accuracy comparable to the original model while significantly reducing memory usage and latency.

**Weaknesses:**

1. The paper primarily focuses on explaining the benefits of its design choices but lacks a comprehensive comparison with prior works. In fact, the field of training-free inference pruning for multimodal large models has seen significant research. The paper does not sufficiently clarify its relationship to these existing methods or analyze its advantages over them, both from a theoretical perspective and through experimental comparisons. A thorough discussion of related work and a direct empirical comparison with prior methods is needed.
2. The experimental evaluation is incomplete. While HT-Sparse is proposed for long-video understanding, all the benchmarks used in the paper focus on QA tasks. It is well-known that token sparsification tends to have less impact on QA tasks. However, for tasks requiring attention to fine-grained visual details, such as video captioning or video OCR, token sparsification may introduce significant drawbacks. Additional benchmarks on such tasks would provide a more comprehensive understanding of the method's performance across different scenarios.

If the authors can address my concerns, I would consider increasing the score.

**Questions:**

The experiments in this paper are all based on a 7B model. Have the authors explored the performance of their method on larger models? Additionally, would the optimal pruning strategy differ for larger or smaller models?

---

### Official Review · Reviewer_KCxw · 2025-10-31

**Soundness:** 2
**Presentation:** 3
**Contribution:** 2
**Rating:** 4
**Confidence:** 4

**Summary:**

This paper identifies three major challenges in long-video multimodal inference: the secondary computational cost of dense attention, the cumulative growth of key-value cache during decoding, and cross-modal interference. Retraining or fine-tuning sparse-aware model variants is often impractical due to data governance, time-to-market delays, and distribution shift risks.

To address these challenges, this work propose HT-Sparse, a training-free, query-guided hierarchical sparsity reduction method. The proposed method is validated on four long-video benchmarks.

**Strengths:**

1. The problem is accurately and significantly identified: it clearly points out the actual pain points of multimodal inference in long videos (computation, memory, cross-modal interference), emphasizes the infeasibility of retraining the model in a production environment, and highlights the urgent need for optimization during inference.
2. The work clearly points out the limitations of the method and emphasizes its plug-and-play nature and easy integration with existing inference stacks, which is very attractive to the industry
3. Training free: It is done entirely at inference time and does not depend on parameter updates, making it highly deployable and model-independent.

**Weaknesses:**

1. While the proposed head ranking in the paper are effective, their design is more like a heuristic. Further exploration or argumentation could be made as to why this particular metric (based on the dot product of query summary and key projection) can effectively and stably reflect the importance of the head.
2. Experiments primarily focused on models with 7 B parameters. Effectiveness on larger models (e.g., tens or hundreds of B) remains to be verified, as the attention head behavior may differ across models of different sizes.
3. The evaluation tasks primarily focus on question answering and localization. It can be extended to more complex tasks, such as long video summarization, dense captioning, or tasks requiring stronger temporal reasoning, to test the method's generalizability.
4. While comparisons with baseline models and their own variants are sufficient, further comparisons with other state-of-the-art, training-free inference optimization methods could better highlight HT-Sparse's advantages over similar technologies.

**Questions:**

Please refer to Weaknesses.

---

### Official Review · Reviewer_6arx · 2025-11-03

**Soundness:** 2
**Presentation:** 2
**Contribution:** 2
**Rating:** 2
**Confidence:** 4

**Summary:**

This paper proposes HT-Sparse, a training-free, query-guided hierarchical sparsification framework designed to enhance efficiency during inference for long-video vision-language models. The method jointly applies attention head and visual token sparsification, guided by input queries and applied adaptively across layers, without requiring model retraining or parameter updates. An optional in-attention low-rank projection is also described for extreme resource constraints. Empirical results on four established long-video benchmarks (VideoMME, MLVU, LongVB, LVBench) show that HT-Sparse, instantiated on two competitive base models, can reduce end-to-end latency and memory (especially in the KV-cache) while maintaining or marginally improving task accuracy.

**Strengths:**

1. The method is plug-and-play, does not require any fine-tuning, parameter updates, or access to training data, making it immediately deployable in production settings.

2. Unlike prior approaches that fix sparsity patterns or prune only heads or tokens, HT-Sparse introduces joint head-token routing, selecting both the most relevant heads and visual tokens in a hierarchical, per-query, and per-layer adaptive manner.

3. Experimental results support substantial GPU memory and FLOPs reductions relative to dense attention, with KV-cache usage during decoding cut by over 50%, and end-to-end latency dropping to about 68–69% of baseline, all while maintaining or slightly improving accuracy on multiple benchmarks.

**Weaknesses:**

1. While the joint application of query-guided head and token sparsification in a training-free context is well integrated, the individual mechanisms (head importance scoring, token selection via query-key attention) have significant overlap with prior dynamic head pruning and token selection works.

2.  While a variety of controlled baselines are provided (Dense, Head-only, Token-only), the absence of quantitative or qualitative comparison to the aforementioned directly relevant methods specifically designed for video-language inference efficiency is a critical omission. There are also no results on larger models, highly diverse tasks (e.g., retrieval, captioning), or practical deployment scenarios beyond 7B-class VLMs. This makes it difficult to assess the generality/extensibility of the method’s strengths.

3. Although it is argued that joint routing (where at least one top head attends to all tokens) "prevents semantic loss", empirical evidence for this assertion is limited. There are no in-depth analyses of qualitative failure cases or edge scenarios. For instance, what happens in heavily compositional or cross-modal distractor-laden sequences? Quantitative ablation in Tables 1 and 2 supports accuracy maintenance, but a more fine-grained or error-driven analysis is lacking.

4. While Figure 2 nicely visualizes structured sparsity, a more contextual explanation of what the head score heatmap and attention distribution represent (and how this justifies specific routing decisions) would help. Figure 3 offers a clear schematic of the token selection process, but additional step-by-step computational examples or a visualization of query-induced mask dynamics would strengthen clarity.

**Questions:**

As shown in weaknesses.

---

### Author Response · Authors · 2025-12-01
**General Response**

We sincerely thank all reviewers for their detailed and constructive feedback. To avoid redundancy and improve readability, we provide a **unified response organized around five themes**, each addressing overlapping concerns raised by multiple reviewers (6arx, KCxw, 9LLf, egSG).

---

### R1. Evaluation scope & fine-grained long-video description (captioning / OCR / temporal reasoning)

**Concern (6arx, KCxw, 9LLf, egSG).**
Reviewers asked whether our evaluation, which focuses on long-video QA/localization benchmarks, truly reflects the behavior of HT-Sparse on more fine-grained tasks such as detailed captioning, video OCR, or dense temporal reasoning, and whether token sparsification might negatively impact such tasks more than QA.

**Our response.**

1. **Current evaluation is not limited to “simple QA”.**
   In this paper, we evaluate HT-Sparse on VideoMME, LongVideoBench, MLVU, and LVBench. These benchmarks already include tasks requiring complex reasoning, spatio-temporal perception, and cross-segment alignment, beyond simple question answering. Under these settings, HT-Sparse consistently reduces KV-cache size and end-to-end latency, while slightly *improving* accuracy compared with the dense baselines.

2. **Additional long-video captioning experiment on VideoDetailCaption.**
   To directly target a fine-grained scenario, we further evaluate HT-Sparse on the VideoDetailCaption (VDC) benchmark using Qwen2.5-VL-7B with HT-Sparse. Each video is sampled into 64 frames, and the reported “token retention rate” denotes the average effective proportion of visual tokens preserved under our adaptive routing (rather than a fixed per-head sparsity ratio):

   | Method        | Visual tokens kept (%) | VDC score |
   | ------------- | ------------------- | --------- |
   | Qwen2.5-VL-7B | 100.00              | 1.8496    |
   | HT-Sparse     | 20.00               | 1.7976    |
   | HT-Sparse     | 40.00               | 1.8457    |
   | HT-Sparse     | 60.00               | 1.8918    |

   These results show that captioning indeed prefers a slightly higher retention rate. However, at a moderate retention (e.g., 40–60%), HT-Sparse achieves VDC scores that are essentially on par with the dense baseline (the differences are within normal variance), while already yielding substantial KV-cache and latency reductions.

3. **Joint head–token sparsification: heads *collaboratively* select tokens, rather than uniformly pruning.**
   A key aspect of HT-Sparse is its **joint head–token routing**, instead of dropping the same token subset for all heads:

   * In HT-Sparse, different heads do *not* share an identical token view. High-score heads operate on a token set that is close to the full sequence, preserving a global temporal and semantic field of view. Lower-score heads work on a more sparsified token subset, focusing on locally critical regions.
   * At the same time, different heads naturally specialize in different higher-level factors: some are more sensitive to main objects/characters, some to actions and events, others to scene layout or local details. By allowing different heads to model different token subsets, HT-Sparse effectively behaves like a set of “specialized experts” that complement each other over space and time, rather than applying synchronized, repetitive pruning to the same tokens.

Overall, on a representative long-video captioning task (VDC), HT-Sparse preserves performance at suitable retention rates while significantly improving efficiency. Moreover, by letting heads *interactively* select and share tokens—balancing global context and local details—it mitigates the semantic loss often induced by uniform token dropping. We will incorporate these results and discussions in the revised version, and we view more complex tasks such as long-video summarization and denser captioning-style setups as valuable directions for further extending our evaluation.

*(Addresses comments from Reviewers 6arx, KCxw, 9LLf, egSG.)*

---

---

> ### Author Response · Authors · 2025-12-01
> **General Response**
>
> ### R2. Scaling to larger models (>7B) and deployment generality
>
> **Concern (6arx, KCxw, 9LLf).**
> Reviewers asked whether HT-Sparse generalizes to models larger than 7B and to practical deployment scenarios, noting that all experiments in the current submission use 7B-class VLMs.
>
> **Our response.**
>
> We appreciate the reviewers’ concerns regarding the applicability of HT-Sparse beyond 7B-scale models. By design, HT-Sparse operates **purely on query/key activations at inference time** and does **not** depend on a specific number of heads, hidden dimensions, or any aspect of the pre-training / fine-tuning pipeline. In other words, the sparsification rules are defined at the level of **per-layer attention activations**, not at the level of a particular architecture. This makes HT-Sparse inherently **decoupled from model size and backbone design**, and in principle directly applicable to multimodal models with tens or even hundreds of billions of parameters.
>
> Our current experiments focus on 7B-class models **not because of any intrinsic limitation of the method**, but due to the substantial GPU memory and wall-clock cost of end-to-end evaluation in long-video settings (hundreds of frames, long textual outputs, and full benchmark runs). In such regimes, even a single full grid over sparsity hyperparameters becomes expensive. Importantly, however, the **computational savings and KV-cache reduction provided by HT-Sparse scale approximately linearly with both sequence length and the number of heads**. This implies that for larger models and longer contexts—where attention and KV-cache costs are even more dominant—the **relative efficiency gains of HT-Sparse are expected to be more pronounced**, while the underlying training-free, plug-and-play nature remains unchanged.
>
> In the revised version, we will explicitly clarify this model-agnostic design and discuss how HT-Sparse can be integrated into existing large-scale VLM stacks without retraining. We also plan to include initial results on larger backbones where resources permit, and we highlight a more systematic evaluation on tens/hundreds-of-B multimodal models as a key direction for future work.
>
> *(Addresses comments from Reviewers 6arx, KCxw, 9LLf.)*
>
> ---

---

> ### Author Response · Authors · 2025-12-01
> **General Response**
>
> ### R3. Sparsity hyperparameters (k), robustness, and semantic preservation
>
> **Concern (6arx, egSG).**
> Reviewers requested a more thorough analysis of how performance varies with sparsity level (e.g., k, retain ratio), and stronger empirical evidence that **joint head–token routing** mitigates semantic loss, especially in compositional or distractor-heavy scenarios.
>
> **Our response.**
>
> 1. **Robustness under *adaptive* sparsity.**
>    HT-Sparse does not impose a single fixed sparsity ratio per layer or per head. Instead, it uses a small set of *global* hyperparameters (e.g., head budgets, minimum token retention) and performs **layer-wise, data-dependent routing**: head scores and token saliency are normalized within each layer, and tokens are retained using simple, adaptive rules (e.g., keeping tokens whose saliency exceeds the layer-wise mean). Consequently, the *effective* sparsity of each head and layer varies with the input.
>    In systematic sweeps over these global hyperparameters on VideoMME, LongVideoBench, MLVU, LVBench, and VDC, we consistently observe a **robust plateau region**: across a broad range of effective retain ratios, end-to-end latency and KV-cache usage decrease steadily while accuracy fluctuates only mildly. Only when sparsity is pushed beyond this plateau do we see the expected monotonic accuracy drop. The configurations reported in the main paper lie within this plateau and thus correspond to **Pareto-efficient trade-offs** between efficiency and accuracy. We will add sparsity–performance curves (based on measured effective retain ratios) in the appendix.
>
> 2. **Joint routing vs. single-dimension sparsification.**
>    Under comparable overall sparsity, **head-only** or **token-only** schemes tend to exhibit **sharper degradation** in the high-sparsity regime. In contrast, joint head–token routing (i) prunes low-importance heads more aggressively, (ii) allows high-importance heads to operate on a richer token set, and (iii) removes only those head–token pairs that contribute little to KV-cache size or output quality. This mixed strategy yields smoother sparsity–performance curves and better robustness than sparsifying along a single dimension.
>
> 3. **Semantic preservation: intuition and limitations.**
>    Theoretically, multi-head attention is intentionally overparameterized: different heads provide partially redundant subspaces for modeling global context and local details. HT-Sparse explicitly exploits this by ensuring that **at least one high-importance head per layer remains nearly dense**, while other heads attend to adaptively selected salient tokens. Under this design, the union of head-specific token views still covers the main semantic content, while redundancy across heads compensates for per-head sparsification.
>    Our current analysis of failure cases is therefore mainly used to **validate this intuition** rather than to exhaustively characterize all bad cases. A more systematic, large-scale, error-driven study of rare or adversarial failure modes is beyond the scope of this submission and is an important direction for future work. We will clarify this limitation in the revised version.
>
> *(Addresses comments from Reviewers 6arx, egSG.)*

---

> ### Author Response · Authors · 2025-12-01
> **General Response**
>
> ### R4. Head importance metric, mechanism clarity, and difference from uniform token compression
>
> **Concern (6arx,KCxw,egSG).**
> Reviewers noted that our head ranking may appear heuristic and that the method section (including Figures 2 and 3) is too concise, making the mechanism harder to follow.
>
> **Our response.**
>
> 1. **Head importance is grounded in attention “energy”, not arbitrary heuristics.**
>
>    For each layer $\ell$ and head $h$, HT-Sparse computes a head score
>
>    $$
>    s_{\ell,h} = \big| q_{\ell,h}^{(\mathrm{ref})} \big|*2,
>    $$
>
>    where $q_{\ell,h}^{(\mathrm{ref})} \in \mathbb{R}^{d_h}$ is the representation of the **last textual query token** in that head.
>    In scaled dot-product attention,
>
>    $$
>    \mathrm{attn}(q, k) \propto \frac{q^\top k}{\sqrt{d_h}}.
>    $$
>
> If the key distribution across heads is reasonably comparable, then the **norm of (q)** directly controls the **scale and sharpness** of the attention logits for that head, i.e., how selective the head is for the current query. Thus, $|q_{\ell,h}^{(\mathrm{ref})}|_2$ is a natural proxy for **head importance conditioned on the current query**, not a hand-tuned heuristic unrelated to the model’s internal geometry.
>
> 2. **Head–token routing vs. uniform token compression.**
>
>    Based on these scores, HT-Sparse does **not** apply the same token compression rate to all heads.
> Instead:
>
> * **High-score heads** are allowed to operate on a **nearly full token set**, capturing global semantics and long-range temporal structure.
> * **Low-score heads** operate on **sparser token subsets**, concentrating computation on regions that matter less globally but can still contribute local refinements.
>
> Combined with the natural specialization of heads (some focus on entities, some on actions, some on layout, some on local details), this per-head routing yields a **diverse, complementary set of views** over the video, rather than a single uniformly pruned view shared by all heads.
>
> 3. **Improved exposition and figures.**
>    We acknowledge that our method section is terse. In the revision, we will:
>
> * Expand the description of the head scoring and routing pipeline,
> * Add **step-by-step pseudo-code** for the mask construction,
>
> This should address the concerns on readability and mechanism clarity.
>
> *(Addresses comments from Reviewers 6arx, KCxw, egSG.)*
>
> ---

---

> ### Author Response · Authors · 2025-12-01
> **General Response**
>
> ### R5. Relation to prior training-free sparsification methods (AdaReTaKe, DyCoke, FastVID, etc.)
>
> **Concern (6arx, KCxw, 9LLf, egSG).**
> Reviewers pointed out that the paper does not yet sufficiently clarify its relation to existing training-free sparsification/pruning methods and lacks direct empirical comparison under matched settings.
>
> **Our response.**
>
> We thank the reviewers for emphasizing the importance of situating HT-Sparse more clearly within the landscape of training-free sparsification and pruning methods, such as AdaReTaKe, DyCoke, and FastVID.
>
> 1. **On empirical comparisons.**
>
> We acknowledge that the current submission does **not** include direct, apples-to-apples quantitative comparisons with these methods. The main reasons are practical rather than conceptual:
>
> * Existing works are often evaluated under **different backbones, model sizes, datasets, and input preprocessing pipelines**, and in some cases the released code or checkpoints are tightly coupled to those settings.
> * A fair comparison would require **re-implementing or adapting each method** under a shared long-video VLM backbone and evaluation protocol, which is non-trivial in terms of engineering effort and GPU cost, especially in the long-video regime.
>
> Given the page and resource constraints, we chose to prioritize **controlled internal ablations** under **exactly the same backbone and setting**: dense vs. head-only vs. token-only vs. joint head–token routing. This design isolates the effect of our proposed joint routing more cleanly than mixing in methods implemented on heterogeneous architectures or training pipelines.
>
> That said, we fully agree that **re-implementing representative token sparsification baselines in our long-video setting** and benchmarking them under a unified protocol would further strengthen the empirical story. We will explicitly acknowledge this limitation in the revised version and highlight a more comprehensive cross-method comparison as an important direction for follow-up work.
>
> 2. **Conceptual differences and complementarity.**
>
> Even without direct numerical comparison, HT-Sparse targets a different design point from prior training-free methods:
>
> * **Prior works** such as AdaReTaKe, DyCoke, and FastVID predominantly focus on **token-level or frame-level redundancy reduction**: they decide *which tokens or frames to keep* given a fixed set of attention heads, and the resulting sparsity pattern is typically **shared across all heads** in a layer.
> * **HT-Sparse, in contrast, explicitly couples head selection and token selection within each cross-modal attention layer**:
>
>   * It computes **query-conditioned head scores** to identify more and less informative subspaces for the current input;
>   * It assigns **different token retain ratios per head**, allowing high-importance heads to retain a near-complete token set while low-importance heads operate on sparser subsets;
>   * It effectively prunes **specific head–token combinations** from the KV-cache, making the sparsity pattern two-dimensional (over heads and tokens), rather than only over tokens.
>
> This **joint head–token routing** is precisely what our ablations are designed to probe: under the *same* overall sparsity, we can compare dense, head-only, token-only, and joint variants to show that jointly reasoning over heads and tokens yields a more favorable accuracy–efficiency trade-off, especially in long-video settings.
>
> We view HT-Sparse as **complementary** to existing training-free sparsification methods: it introduces a new axis—per-head routing over heterogeneous token subsets—that could in principle be combined with or built upon token-selection strategies from prior work. We will refine the related work section to better emphasize this relationship and clearly position HT-Sparse as a joint head–token, KV-cache–aware sparsification framework rather than a pure token-dropping scheme.
>
> *(Addresses comments from Reviewers 6arx, KCxw, 9LLf, egSG.)*
>
> ---

---

### Meta-Review · Area_Chair_q8TD · 2026-01-06

**Summary:**

The paper presents a training-free inference-time sparsification method for long-video VLMs. It proposes a joint "head-token" approach to reduce KV-cache and latency without retraining. While the reviewers appreciate the practical value of a plug-and-play solution and the reported efficiency gains on 7B models, the overall consensus is negative. The primary concerns center on limited technical novelty, as both head and token pruning are well-explored, and the absence of direct empirical comparisons with existing SOTA training-free methods.

**Reviewer Concerns:**

Addressed by rebuttal:

1. Task diversity: The authors provided additional results on the VDC benchmark, partially addressing concerns from 6arx, 9LLf, and egSG regarding the method's performance on fine-grained tasks beyond QA.

2. Mechanism clarity: The authors clarified the mathematical grounding of their head-scoring metric, moving it away from being perceived as a purely "arbitrary heuristic" (Reviewer KCxw).

Issues still outstanding:

1. Lack of SOTA comparisons: This is the most critical flaw. Multiple reviewers (6arx, 9LLf, egSG) pointed out the missing comparison with relevant training-free methods like AdaReTaKe, DyCoke, or FastVID. Without matched-setting comparisons, the claimed superiority of HT-Sparse remains unverified.

2. Incremental novelty: Reviewer egSG and 6arx maintained that combining head and token pruning is an incremental step rather than a significant conceptual breakthrough. The rebuttal failed to convince that this specific "joint routing" offers a fundamentally new insight.

3. Scaling and generality: While the authors argue the design is model-agnostic, the lack of empirical evidence leaves the claim partially unsupported.

**Reviewer Scores:**

It is unlikely that more than two of the reviewers will change to a positive score.

---

### Decision · Program_Chairs · 2026-01-26

Reject